# A *PDCD1* Role in the Genetic Predisposition to NAFLD-HCC?

**DOI:** 10.3390/cancers13061412

**Published:** 2021-03-19

**Authors:** Nardeen Eldafashi, Rebecca Darlay, Ruchi Shukla, Misti Vanette McCain, Robyn Watson, Yang Lin Liu, Nikki McStraw, Moustafa Fathy, Michael Atef Fawzy, Marco Y. W. Zaki, Ann K. Daly, João P. Maurício, Alastair D. Burt, Beate Haugk, Heather J. Cordell, Cristiana Bianco, Jean-François Dufour, Luca Valenti, Quentin M. Anstee, Helen L. Reeves

**Affiliations:** 1Translational and Clinical Research Institute, Faculty of Medical Sciences, The Medical School, Newcastle University, Framlington Place, Newcastle upon Tyne NE2 4HH, UK; nardeen.rafat@mu.edu.eg (N.E.); Misti.McCain@newcastle.ac.uk (M.V.M.); Robyn.Watson@newcastle.ac.uk (R.W.); ejayliu@hotmail.com (Y.L.L.); marcozaki@mu.edu.eg (M.Y.W.Z.); a.k.daly@newcastle.ac.uk (A.K.D.); J.Mauricio2@newcastle.ac.uk (J.P.M.); alastair.burt@newcastle.ac.uk (A.D.B.); quentin.anstee@newcastle.ac.uk (Q.M.A.); 2Biochemistry Department, Faculty of Pharmacy, Minia University, Minia 61519, Egypt; mostafa_fathe@minia.edu.eg (M.F.); michael.fawzy777@mu.edu.eg (M.A.F.); 3Population Health Sciences Institute, Faculty of Medical Sciences, Newcastle University, International Centre for Life, Newcastle upon Tyne NE1 3BZ, UK; rebecca.darlay@newcastle.ac.uk (R.D.); heather.cordell@newcastle.ac.uk (H.J.C.); 4Biosciences Institute, Faculty of Medical Sciences, The Medical School, Newcastle University, Framlington Place, Newcastle upon Tyne NE2 4HH, UK; Ruchi.Shukla@newcastle.ac.uk (R.S.); nikkimartinamcstraw@hotmail.co.uk (N.M.); 5Department of Cellular Pathology, Royal Victoria Infirmary, Newcastle upon Tyne Hospitals NHS Foundation Trust, Newcastle NE1 4LP, UK; beate.haugk@nhs.net; 6Translational Medicine, Department of Transfusion Medicine and Hematology, Fondazione IRCCS Cà Granda Ospedale Maggiore Policlinico, 20122 Milan, Italy; cristiana.bianco@policlinico.mi.it (C.B.); luca.valenti@unimi.it (L.V.); 7University Clinic for Visceral Surgery and Medicine, University Hospital of Bern, 3010 Bern, Switzerland; jf.dufour@svmed.ch; 8Hepatology, Department of Biomedical Research, University of Bern, 3012 Bern, Switzerland; 9Department of Pathophysiology and Transplantation, Università degli Studi di Milano, Fondazione IRCCS Cà Granda Ospedale Maggiore Policlinico, 20122 Milan, Italy; 10The Liver Unit, Freeman Hospital, Freeman Road, Newcastle upon Tyne Hospitals NHS Foundation Trust, Heaton NE7 7DN, UK

**Keywords:** primary liver cancer, hepatocellular carcinoma, metabolic syndrome, genetic predisposition, single-nucleotide polymorphism, *PDCD1*, PD-1, *PNPLA3*, *TM6SF2*

## Abstract

**Simple Summary:**

Many more people are dying each year from primary liver cancers arising in obesity-related fatty liver disease. Often these cancers are a consequence of fatty liver disease progression, with inflammation, scarring and cirrhosis. Less often, cancers develop in the presence of fat without cirrhosis. Evidence from animal models suggests the immune response to fat is important. We have explored genetic variations in candidate immunoregulatory genes. Our study of nearly one-thousand patients with fatty liver disease, comparing 391 with cancers to 594 without, indicates that genetic variation in a gene (*PDCD1*) that codes for the T cell receptor PD-1 may be important. Inherited variations that affect function of immunoregulatory proteins like PD-1 may underpin why some patients with fatty liver disease—whether they have cirrhosis or not—are more likely to develop liver cancer.

**Abstract:**

Obesity and non-alcoholic fatty liver disease (NAFLD) are contributing to the global rise in deaths from hepatocellular carcinoma (HCC). The pathogenesis of NAFLD-HCC is not well understood. The severity of hepatic steatosis, steatohepatitis and fibrosis are key pathogenic mechanisms, but animal studies suggest altered immune responses are also involved. Genetic studies have so far highlighted a major role of gene variants promoting fat deposition in the liver (*PNPLA3* rs738409; *TM6SF2* rs58542926). Here, we have considered single-nucleotide polymorphisms (SNPs) in candidate immunoregulatory genes (*MICA* rs2596542; *CD44* rs187115; *PDCD1* rs7421861 and rs10204525), in 594 patients with NAFLD and 391 with NAFLD-HCC, from three European centres. Associations between age, body mass index, diabetes, cirrhosis and SNPs with HCC development were explored. *PNPLA3* and *TM6SF2* SNPs were associated with both progression to cirrhosis and NAFLD-HCC development, while *PDCD1* SNPs were specifically associated with NAFLD-HCC risk, regardless of cirrhosis. *PDCD1* rs7421861 was independently associated with NAFLD-HCC development, while *PDCD1* rs10204525 acquired significance after adjusting for other risks, being most notable in the smaller numbers of women with NAFLD-HCC. The study highlights the potential impact of inter individual variation in immune tolerance induction in patients with NAFLD, both in the presence and absence of cirrhosis.

## 1. Introduction

Commonly arising on a background of cirrhosis, the geographic variations in global incidence and mortality attributed to hepatocellular carcinoma (HCC) have largely reflected differences in the prevalence of chronic viral hepatitis, namely, hepatitis B (HBV) and hepatitis C (HCV) [1]. Alcohol-related liver disease (ARLD) has also been recognised as a significant contributor to risk, particularly in Western nations [1]. With over hundreds of thousands of new cases and deaths per year, for a cancer where the major risk factors are known and many cases potentially preventable or treatable if detected at an early stage, the needs to better understand this and exploit opportunities to intervene and improve outcomes are widely recognised. In some countries with a particularly high incidence of HCC, population-level interventions have had an impact, with reductions in HCC mortality consequent to HBV immunisation [2] or early HCV-HCC detection and treatment [3]. The introduction of effective antiviral therapies for both HBV and HCV have also heralded major scientific advances [4,5]. It is disappointing, therefore—that in contrast to the majority of other cancers—HCC deaths globally have continued to increase [6].

Epidemiological studies in the Western nations have highlighted a close association between HCC and the rising prevalence of obesity, associated with the metabolic syndrome [7]. Non-alcoholic fatty liver disease (NAFLD) is the liver manifestation of this syndrome, and while primary care management targeting diabetes and hypertension [8], as well as public health campaigns to reduce smoking [9], have had a significant impact on cardiovascular deaths, the prevalence, morbidity and mortality consequent to NAFLD and NAFLD-HCC have escalated. NAFLD is estimated to affect 25% of Western adults [10]. Although the majority of those have steatosis without significant inflammation and fibrosis, the prevalence is so high that even with only a minority of patients developing progressive disease, NAFLD is now one of the commonest causes of cirrhosis and HCC [7]. In patients with known NAFLD cirrhosis who are fit for therapeutic intervention, HCC surveillance with bi-annual liver ultrasound is widely endorsed, although views of its cost effectiveness and benefit are frequently debated [11]. One of the consequences of ineffective or failed surveillance is that NAFLD-HCC is more often detected at an advanced stage, where older age and metabolic syndrome associated comorbidities further limit treatment options. There is a pressing need to develop a better understanding of why some patients with NAFLD develop HCC—both in the presence or absence of cirrhosis—if we are to identify clinically useful biomarkers that will inform intervention strategies or more cost-effective approaches to surveillance and early detection.

Previous studies have recognised that NAFLD progression to advanced fibrosis or cirrhosis is associated with diabetes, overweight, male sex and advanced age, as well as a genetic predisposition [12,13]. Although there have not been genome wide association studies (GWAS) reporting genetic variables associated with NAFLD-HCC risk, single-nucleotide polymorphisms (SNPs) in a number of candidate genes associated with NAFLD development have also been studied in the context of NAFLD-HCC. SNPs in genes that promote fat accumulation in hepatocytes as well as reportedly promoting NAFLD-HCC include rs738409 in patatin-like phospholipase domain-containing 3 (*PNPLA3*) [13,14] and rs58542926 in transmembrane-6 superfamily member-2 (*TM6SF2*) [7,15,16,17]. The sensitivities and specificities of individual SNPs such as these for determining HCC risk in NAFLD is poor. However, their clinical utility may be improved using combinations of risk associated SNPs. A polygenic risk score (PRS) combining genetic variations in *PNPLA3* and *TM6SF2*, with others in membrane bound O-acyltransferase domain containing 7 (*MBOAT7*), glucokinase regulator (*GCKR*) and 17β-hydroxysteroid dehydrogenase type 13 (*HSD17B13*), has recently been proposed [18].

While increased liver fat stores appear to increase the risk of HCC development [18], data from animal models and human studies have also highlighted a role of the immune environment in the modulation of cancer risk in NAFLD [7]. Here, in the hope of enhancing our understanding of NAFLD-HCC in a clinically translatable fashion, we have explored SNPs in a number of candidate HCC immunoregulatory genes, in addition to those in *PNPLA3* and *TM6SF2*, in patients with NAFLD and NAFLD-HCC. The SNPs selected were based on previous reports and included rs2596542 in Major Histocompatibility Complex class I polypeptide related sequence A (*MICA*) [19], rs187115 in Cluster of differentiation 44 (*CD44*) [20,21,22], as well as rs7421861 and rs10204525 in programmed cell death-1 (*PDCD1*) which encodes PD-1 [23,24,25,26,27,28,29,30,31,32,33]. *PDCD1* rs10204525 C < T is common in Asian populations, with functional implications for patients with HBV. The data presented here highlight genetic variability in *PDCD1* associated with HCC risk in NAFLD in European patients. The clinical implications are discussed.

## 2. Materials and Methods

### 2.1. Patients

This study included a total of 594 NAFLD Caucasian patients derived from the European NAFLD Registry [34] who were considered as controls, and 391 patients with HCC attributed to NAFLD. The primary cohort was from the Newcastle upon Tyne Hospitals NHS Foundation Trust, Newcastle, UK (controls 416, HCC 198). Recruitment of patients with HCC was from 1 January 2004 until 3 December 2019, with a minimum follow-up period of 12 months, until 31 December 2020. There were two replication cohorts, from Inselspital Hospital, Bern, Switzerland (76 controls, 84 HCC), and Milan, Italy (102 controls with severe fibrosis, 109 HCC), as approved by each institution. All patients signed a consent form. The Berne patients have been recruited since 2010, with follow up ongoing until death or transplantation. For the Milan patients, those with cancer were recruited between 2010 and 2016. The Milan control patients were those with NAFLD and severe fibrosis or cirrhosis, recruited between 2018 and 2020. There was no subsequent follow up of the Milan patients for research purposes. Demographic data included age and sex, with body mass index (BMI) and presence of type 2 diabetes (T2DM) recorded. Each of the controls had a documented history of NAFLD, with an otherwise negative liver screen (HBV and HCV serology, serum ferritin, autoantibody screen, and alpha-1-antitrypsin level). Patients classed as having NAFLD-HCC were men or women with evidence of a fatty liver on biopsy or imaging or having the presence of at least T2DM or BMI > 30, with an otherwise negative liver screen, drinking <21 or 14 units of alcohol per week, respectively, for at least 5 years prior to their first presentation with liver disease. The diagnosis of HCC was established through noninvasive assessment or histologically according to EASL clinical practice guidelines [1].

### 2.2. Genotyping Strategy

DNA was prepared as previously described [13]. SNPs in the candidate genes (*PNPLA3* rs738409, *TM6SF2* rs58542926, *MICA* rs2596542, *CD44* rs187115, *PD-1* rs7421861 and rs10204525), were genotyped by taqman assay (#4351379, Applied Biosystems Inc., Waltham, MA, USA) according to the manufacturers protocol or determined by GWAS as reported previously [35]. Taqman assay details are summarised in Appendix A.

### 2.3. Statistical Analysis

Associations between SNPs and HCC status were analysed by logistic regression in PLINK v1.9 [36], with age, sex, cirrhosis and diabetes status included as additional covariates. Multivariate analyses were expressed as odds ratio (OR) with 95% confidence intervals (CI). Logistic regression was repeated conditioned on the genotypes of both rs738409 and rs58542926.A subset of non-cirrhotic individuals were analysed separately by similar methods. A fixed-effect meta-analysis was performed with METAv1.7 [37] on Newcastle, Berna and Milan cohorts using the log Odds Ratios (lnORs) and standard errors from the individual cohort analyses. Haplotype analysis for *PDCD1* SNPs was carried out using software UNPHASED, as previously described [38]. In the patients with liver cancer, associations were also explored with clinicopathological variables and survival, by Kaplan Meier and Cox regression analyses using SPSS version 25 (IBM Corp., Armonk, NY, USA) licensed to Newcastle University.

### 2.4. Power Calculation

A power calculation of sample size to evaluate the effect of genetic variation on HCC risk in NAFLD was performed using the “case–control for discrete traits” option in GPC software (http://zzz.bwh.harvard.edu/gpc/ (accessed on 3 March 2021)). To perform this, the prevalence of NAFLD-HCC in a NAFLD population similar to our own was estimated as 0.5%, based epidemiological studies data [39], but also from HCC incidence as reported in large cohorts included in recent therapeutic trials for NASH [40,41]. The minor allele frequencies used were those of our own NAFLD control population. These power calculations are summarised in Appendix A. In the combined cohorts, there was an estimated 80% power to detect a significant allelic relative risk of ≥1.3, falling to 55–60% in the Newcastle cohort alone. At a level of relative risk ≥ 1.4, the power was 95% in our combined cohort and 76–80% in the Newcastle cohort.

Graphical abstract was designed using BioRender software (https://biorender.com/ accessed on 2 February 2021; Toronto, ON, Canada).

## 3. Results

### 3.1. The Newcastle Patient Cohorts Clinical Information

#### 3.1.1. The NAFLD Control and NAFLD-HCC Patient Characteristics

The NAFLD control patients included all patients, regardless of age, sex or stage of liver disease, attending the NAFLD specialist clinic in Newcastle who consented to take part in clinical research studies. The cohort was representative of the patients seen. The patients with HCC were also representative, again including all patients who consented to the use of their tissues and data for research. The progression of NAFLD to HCC in Newcastle is known to be associated with increasing age, male sex, T2DM and the presence of cirrhosis—each of which were highly significantly different between the two groups, as summarised in Table 1. Notably, although increasing BMI is well established as a risk factor for NAFLD progression to cirrhosis, BMI was significantly reduced in NAFLD-HCC cases compared to NAFLD controls. The lower BMI may reflect older age and/or sarcopenia, or be a consequence of cancer development, rather than being a predisposing feature—falling particularly in those with more advanced TNM stage (Appendix A). Note, however, that the BMI was lower across all TNM stages relative to controls, particularly in the non-cirrhotic (NC) cases (Appendix A).

#### 3.1.2. The Newcastle NAFLD-HCC Cohort, Comparing Those with and without Cirrhosis

The features of the NAFLD-HCC cases are summarised in Appendix A. In 61.1%, HCC arose in the presence of cirrhosis. As previously reported, the median age of individuals with NC NAFLD-HCC was significantly greater [42]. Tumours were also more advanced in terms of size, with more patients presenting with TNM stage III or IV disease. Notably though, 48% of patients with NC NAFLD-HCC presented with a single large tumour, with only 6.5% classed as TNM Stage II, compared to 35.5% and 36.4% for cirrhotic NAFLD-HCC cases, respectively. These differences in TNM stage were highly significant. Furthermore, the NC NAFLD-HCC cases had relatively preserved liver function, as shown by a number of the parameters (albumin, bilirubin, ascites, encephalopathy, prothrombin time and Childs Pugh Stage) presented in Appendix A. Consequently, despite their advanced age and 36.4% receiving only supportive care, NC NAFLD-HCC patients were twice as likely to be treated surgically. Survival in NAFLD-HCC cases was mostly determined by stage (tumour number, size and portal vein invasion) at presentation rather than the presence of cirrhosis, as summarised in Appendix A. Age, ECOG PST and tumour differentiation were also significant in multivariate Cox Regression analyses.

Raised BMI and T2DM were expectedly common in the NAFLD-HCC cohort overall. The lower BMI in NC NAFLD-HCC cases has been mentioned above, but also of note, T2DM was less prevalent, being present in 59.7% of NC cases compared to 76.0% of cirrhotic cases. While these features are widely associated with cancer risk, it is perhaps not so surprising that they were less striking in the NC NAFLD-HCC cases, given these are also strongly associated with progression to cirrhosis, and by definition the NC cases do not have cirrhosis. The differences do suggest, however, that the key genes and/or pathways in obesity and T2DM promoting NAFLD progression to cirrhosis, versus those promoting NAFLD progression to HCC, are not necessarily the same. The likelihood that other pro-carcinogenic pathways may play a relatively larger role in NAFLD patients without cirrhosis has been previously suggested [43].

The diagnosis of HCC was confirmed histologically in 93/198 cases (47.0%). Histopathology features were often heterogenous even within the biopsies, but the more common architectural type, as well as any subtype described [44], was attributed where possible (*n* = 90). Apart from fibrolamellar cases being restricted to NC only, the distribution of other subtypes was similar in cirrhotic versus NC cancers. Trabecular architecture was common, followed by solid/compact tumours. In some cases, features of both hepatocellular and cholangiocarcinoma were evident, and these were classed as mixed tumours. In over 60% no subtype was reported. In those with a subtype noted, steatohepatitic HCC (SH-HCC) was the most common, present in 20–25% of cases regardless of cirrhosis. In this series, there were no cases described as macrotrabecular massive, chromophobe or neutrophil-rich.

### 3.2. The Newcastle Patients’ Cohort Genotype Data

#### 3.2.1. PNPLA3 rs738409 C > G and TM6SF2 rs58542926 C > T Genotyping Data

All of the genotypes—for fat and immunoregulatory genes—were in Hardy–Wienberg equilibrium in the Newcastle cases and controls cohort. *PNPLA3* rs738409 and *TM6SF2* rs58542926 are two well-studied SNPs where the minor alleles are reported to be associated with the development of HCC in fatty liver attributed to NAFLD or ARLD [13,45]. The allele frequency data for these two SNPs in the Newcastle NAFLD and NAFLD-HCC cohorts is shown in the first two column clusters of Figure 1 and Appendix A. In Europe, 78% of people carry at least one copy of the *PNPLA3* rs738409 “C” allele (https://www.ncbi.nlm.nih.gov/ (accessed on 18 January 2021)), with just 5% being homozygous for the G allele. Figure 1 shows that the percentage of patients carrying the C allele falls quite markedly in patients with NAFLD, with and without HCC in Newcastle. Notably, >20% of those with NAFLD-HCC were homozygous for the variant G allele. The differences in *PNPLA3* genotype assessed by logistic regression were significant, with statistical tests assessing dominant and recessive models of inheritance shown in Appendix A. For *TM6SF2* rs58542926 (yellow bars Figure 1), both individuals heterozygous or homozygous for the variant T allele were significantly elevated in patients with NAFLD-HCC, although—as expected—the minor allele frequency *TM6SF2* rs58542926 was lower than that of *PNPLA3*.

Figure 1 The figure shows the allelic frequency (wild type, heterozygous mutant, homozygous mutant) of different candidate SNPs in fat-regulatory genes (*PNPLA3* and *TM6SF2*) and immune-regulatory genes (*CD44*, *MICA* and *PDCD1*) in NAFLD (*n* = 416; solid bars) and NAFLD-HCC (*n* = 198; dotted bars) patients from Newcastle cohort.

#### 3.2.2. The Newcastle Patients’ Cohort Genotype Data—Immunoregulatory Genes

There was no evidence that any of the candidate immunoregulatory genes were associated with the development of NAFLD, when comparing the minor allele frequency (MAF) to controls from the European NAFLD GWAS [34]. The MAF for each of the SNPs (bar rs10204525 which failed quality control) are shown in Appendix A. There were no statistically significant differences in genotypes for either *CD44* rs187115 or *MICA* rs2596542 when comparing the NAFLD versus NAFLD-HCC cases (grey bars Figure 1; Appendix A). Considering the SNPs tested in *PDCD1* however, there was quite a difference in allele frequencies for rs7421861 A > G, with the wild type A allele being significantly more common in patients with NAFLD-HCC. This was in keeping with the variant allele protecting against the development of HCC in NAFLD, as the wild type A allele was more common in the patients with cancer. The *PDCD1* rs10204525 T allele was not common and was not significantly different in NAFLD-HCC versus NAFLD controls.

#### 3.2.3. The Newcastle Cohort-Multivariate Analyses Exploring Associations with NAFLD-HCC

For each of the fat or immunoregulatory SNPs, we went on to explore their association with the development of HCC in NAFLD, independently of the other established risk factors for NAFLD-HCC—namely, age, gender, the presence of T2DM and the presence of cirrhosis. These data are summarised in Table 2. Of note, while both *PNPLA3* and *TM6SF2* variants are widely acknowledged as promoting NAFLD-HCC, neither was independently significant of all the other risk factors in this single centre cohort. The proposed association of *PDCD1* rs7421861, with the variant allele less common in NAFLD-HCC, remained highly significant when including each of the other risk factors. Notably, the *PDCD1* rs10204525 C > T SNP, acquired significance after adjusting for the other risk factors. Inspection of the data (Appendix A) indicated that the *PDCD1* rs10204525 T allele was more common in Newcastle cohort females with NAFLD-HCC (15/41; 36.6%) compared to female NAFLD controls (30/184; 16.8%, *p* = 0.003, Chi Square).

Considering the possibility of a risk *PDCD1* haplotype, Linkage Disequilibrium analyses between the two *PDCD1* SNPs revealed a D’ value of 1 an R^2^ of 0.0487329. The low R^2^ value reflects the low frequency of the rs10204525 T allele (0.1) relative to the rs7421861 A allele (0.7). However, when the rarer rs10204525 T allele was present, it was always inherited with the more common rs7421861 A allele. These data and haplotype analyses are summarised in Appendix A, supporting the *PDCD1* rs7421861 “A” allele as being the one associated with greatest risk. It would appear, although numbers are small, that the relatively minor contribution of the *PDCD1* rs10204525 T allele, always inherited with the rs7421861 A allele, may be more relevant in women in the Newcastle cohort.

We went on to condition the analyses of the immunoregulatory SNPs on the *PNPLA3* and *TM6SF2* genotypes. The analyses yielded similar results, also shown in Table 2. In the conditioned univariate analysis, the protective effect of *PDCD1* rs7421861 minor allele remained strikingly significant, after adjusting for age, sex, cirrhosis and T2DM. For *PDCD1* rs10204525, where the presence of the variant allele acquired significance after adjusting for age, sex, cirrhosis and T2DM, the odds ratio and significance after conditioning on the fat regulatory genes was enhanced.

#### 3.2.4. The Newcastle Cohort—Genotype Associations within the Cirrhotic Versus Non-Cirrhotic NAFLD-HCC Cases

Differences in genotype associations within cases classed as cirrhotic or NC were explored, and these are summarised in Table 3. Although the *PNPLA3* rs738409 variant G allele is undoubtedly more frequent in cancers compared to controls, there was also a significant difference between the cirrhotic versus the non-cirrhotic cases with NAFLD-HCC (red text Table 3, *p* < 0.0001). As *PNPLA3* rs738409 is associated with advanced NAFLD [35], this was not so surprising. Viewing the groups with the non cancer cases also broken down into those with and without cirrhosis confirmed that the most striking associations of both the *PNPLA3* and the *TM6SF2* variants were with the presence of cirrhosis, rather than with the presence of cancer. As previously reported, the HCC risk associated with these two SNPs is largely mediated by fibrosis, although in larger cohorts it is possible to discern an impact partially independent of fibrosis [18]. In contrast, the prevalence of the *PDCD1* alleles associated with elevated cancer risk (rs7421861 A allele and *PDCD1* rs10204525 T allele, green text Table 3) was clearly associated with HCC, in both the presence and absence of cirrhosis.

We went on to analyse genotypes in a subgroup restricted to controls and cases without cirrhosis. These analyses, including 353 NC NAFLD controls and 78 NC NAFLD-HCC patients, are summarised in Appendix A. In the NC cases, the *PDCD1*rs7421861 G allele was again protective against HCC. Furthermore, significance was retained after including age, gender and T2DM by regression, as well as after conditioning on *PNPLA3* and *TM6SF2*.

There were no overarching genotype associations with any particular architectural or histological subtype of HCC, although there were notable subtype associations within the NC HCC cases. Of 12 classed as SH-HCC, nine were homozygous and three heterozygous for the *PDCD1* rs7421861 “A” allele. Notably, 11/12 also carried the *CD44* rs187115 C allele. The clear cell subtype was rare, but both cases were homozygous for these named *PDCD1* and *CD44* alleles. The findings were statistically significant (*PDCD1* rs742861 *p* = 0.045; *CD44* rs187115 *p*-0.009; Chi Square Exact test). While interesting, much larger numbers of cases with tumour histology would need to be systematically studied to attribute robust biological relevance to these observations.

### 3.3. Berne and Milan NAFLD and NAFLD-HCC Cohorts

Focusing on the predisposition to HCC development in patients with NAFLD, we went on to explore the fat and immunoregulatory SNPs in two additional European cohorts of control NAFLD and NAFLD-HCC cases, from Berne in Switzerland and Milan in Italy. The demographic and NAFLD relevant clinical data for the controls versus patients with HCC are presented in Table 4. In these smaller cohorts, not all the variables were significantly different. Although the trends were similar to the larger Newcastle cohort, there were some differences to note. Compared to Newcastle, the Berne control cases were similar, but the Berne NAFLD-HCC cases were younger, fewer were women and fewer had T2DM. The Milan control cohort was older, compared to Newcastle, with a lower BMI and a much greater prevalence of cirrhosis. The NAFLD-HCC Milan cohort was similar to that of Berne, with younger patients, with a lower BMI, less T2DM and a higher prevalence of cirrhosis.

Compared to Newcastle, the differences in the Milan cohort were at least in part due to differences in selection criteria, as these were patients recruited for a case–control study, as previously described [43]. The patients with NAFLD-HCC were representative of cases seen in that hospital between 2010 and 2016, while the NAFLD controls were not—these were recruited based on the presence of advanced fibrosis or cirrhosis in 2018–2020. The older age and lower BMI in the Milan controls may also be reflective of that selection bias. The Berne patients have been recruited from 2010, with both cases and controls being reflective of patients referred to that hospital. The differences in the cancer patients between the three institutions may reflect country differences. However, the largest cohort in Newcastle was recruited at a tertiary centre where all the patients regardless of age, fitness or stage of cancer were referred, from a large catchment area in the northeast of England. The cohorts from Berne and Milan may not have been as inclusive, potentially influenced by local referral practices (younger, fitter, with less diabetes and less obesity). As the Berne and Milan cohorts were relatively small compared to Newcastle, the differences were noted rather than interpreted further.

In the Berne and Milan cohorts, however, none of the allele frequencies—for either fat regulatory or immunoregulatory SNPs—was significantly different comparing NAFLD controls and NAFLD-HCC cases. The genotype frequencies are shown in Appendix A, with statistical analysis in Appendix A.

### 3.4. Newcastle, Berne and Milan NAFLD and NAFLD-HCC Cohorts Combined

The combined cohort data is summarised in Table 5. The meta-analysis for the *PNPLA3*, *TM6SF2* and the *PDCD-1* SNPs, also conditioned on *PNPLA3* and *TM6SF2* is shown in Table 6.

By univariate analysis, the SNPs in *PNPLA3*, *TM6SF2* and *PDCD1* rs7421861 were significantly different in NAFLD-HCC versus controls, but none retained significance after regression analysis including age, sex and cirrhosis. The *PDCD1* rs10204525 C > T variant again acquired significance in the regression analysis, including after conditioning on *PNPLA3* and *TM6SF2*.

### 3.5. Exploration of Functional Roles for PDCD1 rs7421861 and PDCD1 rs10204525

We have evaluated global effects of these genetic variants on *PD-1* expression by mining publicly available tissues data in the Ensembl Genome (https://www.ensembl.org/index.html (accessed on 10 March 2021)). We used multi-tissue expression quantitative trait locus (eQTL) mapping as a means of exploring whether the SNPs associate with altered gene expression in a given tissue [46,47,48,49]. The potential impact of the SNPs on transcript splicing was also evaluated by mining data from the GTEx portal (https://www.gtexportal.org/home/ (accessed on 10 March 2021) version 8), another publicly available resource. Splicing quantitative trait loci (sQTLs) are assigned for SNPs that are associated with altered or alternative splicing (AS), identified by Leafcutter [50] in the GTEx consortium project.

#### 3.5.1. Elevated PD-1 Expression in the Presence of the PDCD1 rs7421861 “A” Major Allele

In Ensembl, there were 2765 data entries showing gene expression correlations with this *PDCD1* rs7421861, in a variety of tissues. We have assessed the effect of the variant “G” allele on the eQTL for *PD-1* expression, reported relative to “A” as the reference allele. Positive values indicate upregulation of the transcript in presence of the minor allele, while negative values indicate downregulation [51]. There were 357 changes significant at a *p* value < 0.05. Restricting analysis to 84 data entries considering expression in immune cells, blood or spleen, *PD-1* was significantly downregulated in the presence of the minor allele (Table 7). Expression in the presence of the wild type *PDCD1* rs7421861 “A” allele, the one we have identified as associated with cancer risk, was therefore higher, with the minor allele associated with significant suppression. Note that besides *PD-1* expression, *PDCD1* rs7421861 was associated with altered expression of 5 other genes in locus vicinity (restricting analysis to effect size of ≥0.5 or ≤−0.5, Appendix A).

#### 3.5.2. Elevated PD-1 Expression in the Presence of the PDCD1 rs10204525 Minor Allele

Similar to *PDCD1* rs7421861, there were 2652 gene expression data entries enabling the study of correlations with the *PDCD1* rs10204525 variant. Again, restricting the analysis to those focused on expression in immune cells, blood or spleen, with an effect size of ≥0.5 or ≤−0.5; *p* < 0.05, there were 66 data entries. The SNP reportedly acts as a potential cis-eQTL for 11 genes within the locus (Appendix A). Overall, gene expression was upregulated in the presence of the minor allele *PDCD1* rs10204525 “T” allele, the one we have identified as associated with NAFLD-HCC risk, most notably in activated T cells and monocytes (Table 8).

#### 3.5.3. sQTLs Associated with PDCD1 rs7421861 and PDCD1 rs10204525

In the GTEx portal, *PDCD1* rs7421861 and rs10204525 were also identified as having a significant association with altered splicing, measured by the sQTL in different tissues. In particular, *PDCD1* rs7421861 was significantly associated with an altered intron excision ratio for long non-coding RNA LINC01238 in splenic tissue (NES = −0.78, *p* value = 3.3 × 10^−7^), while *PDCD1* rs10204525 was significantly associated with an altered intron excision ratio for LINC01237 in whole blood (NES = 0.47, *p* value = 1.1 × 10^−7^) (Figure 2). Using the LINCs as candidate biomarkers of altered splicing in the region of the *PDCD1* locus, the data point towards the SNPs altering splicing factor binding sites.

## 4. Discussion

As yet, there has been no GWAS study defining the genetic determinants of HCC arising in NAFLD, in either the presence or absence of cirrhosis. Of the NAFLD-HCC-associated candidate genes studied, the most robustly validated thus far have been those identified in GWAS studies looking for genes associated with NAFLD development. It is not a surprise, therefore, that these are variants in genes—like *PNPLA3* and *TM6SF2*—where the minor alleles alter protein function and promote steatosis, associated with both qualitative and quantitative alterations in hepatic fat content [52]. Although their impact on HCC predisposition is directly correlated with the promotion of NAFLD [18], these causal variants involved in fatty liver disease predisposition may also have pleiotropic effects, whereby they directly promote hepatic carcinogenesis by activating specific pathways. However, GWAS designed to identify genes elevating the risk of NAFLD would not necessarily capture those specifically promoting NAFLD-HCC. The candidate immunoregulatory genes studied here were purposefully selected.

A GWAS considering HCC risk in HCV previously identified a candidate immunoregulatory SNP in Major Histocompatibility Complex class I polypeptide related sequence A (MICA) [19]. MICA is a stress-induced cell surface antigen presenting peptide that binds and activates the NKG2D stimulatory receptor on the surface of NK cells and T cells, mediating their cytolytic immune surveillance. It has been reported that *MICA* rs2596542 C > T alters the expression of MICA, decreasing its membrane-bound form and increasing its soluble concentrations [53], favouring HCV progression to HCC in a Japanese population [19]. In our European NAFLD patients studied here, there was no significant difference between NAFLD control and NAFLD-HCC patients. This possibly supports *MICA* variation influencing the immune response and progression to fibrosis in HCV, as recently reported, rather than hepatocarcinogenesis [54].

Cluster of differentiation 44 (CD44) is a multifunctional transmembrane receptor expressed on various cell types, with ligand binding activating intracellular signalling pathways to stimulate cytokine release and lymphocyte activation [20]. The *CD44*rs187115 T > C SNP is located within intron 1, with the C allele reportedly associated with a higher risk of HBV-HCC development in Taiwan and China [21,55]. The mechanisms are yet to be determined, but a role in the p53 stress response has been proposed [22]. Our own interest in *CD44* arose having identified its expression in hepatic macrophages, in association with NC SH-HCC in murine models of NAFLD-HCC [56]. Although there was no significant differences in allele frequencies between NAFLD controls and NAFLD-HCC cases, the C allele was noted to be significantly more common in SH-HCC arising in NC NAFLD. This was interesting, but the numbers of cases studied are presently small.

*PDCD1* encodes PD-1, a co-inhibitory receptor expressed most notably on activated T cells, with binding to the widely expressed ligands PD-L1 and PD-L2 suppressing activity and limiting potential damage to the host [24]. T cell receptor (TCR) activation stimulates the expression of nuclear factor of activated T cells (NFAT), a transcription factor that promotes IL-2-mediated T cell proliferation, but also binds the *PDCD1* promoter inducing expression of PD-1 [23]. Ongoing T cell activation with persistent PD-1 expression is associated with functional T cell exhaustion, with loss of T cell function and enhanced apoptosis. Tumour overexpression of PD-L1 is recognised as a means of supporting an immunosuppressive tolerant microenvironment and tumour immune evasion [57]. Sustained PD-1 expression, its decreased degradation, or expression of PD-L1 may also reportedly elevate individuals susceptible to HCC development [24,25,58,59].

A number of SNPs have been identified that may alter the expression and function of PD-1, with reports linking these to autoimmune disease [59], but also immunosuppressive conditions. *PDCD1* rs10204525 C > T is located at the 3′UTR, increasing PD-1 expression and reportedly promoting dysfunctional T cell responses and persistence in HBV infection [26,27,28,29,30,31,33,60]. The promotion of an immunosuppressive phenotype by *PDCD1* rs10204525 C > T may also contribute to tumour development but has not as yet been characterised. Our in silico analysis an influence of *PDCD1* rs10204525 on *PD-1* expression, being higher in the presence of the *PDCD1* rs10204525 “T” allele, with evidence in addition for a regulatory role in splicing. In our study, after adjusting for other cancer risks including gender, the variant “T” allele was significantly associated with NAFLD-HCC risk, regardless of the presence or absence of cirrhosis. As the T allele has low prevalence in European populations (~10%), the significance became much more noticeable in our cases after adjusting for gender—as over 35% of women with NAFLD-HCC carried the variant. Notably, the T allele is in fact the major allele in Asian populations, being present in 60–70% of people (NCBI dbSNP).

The rs7421861 A > G, the second *PDCD1*candidate SNP studied, is located in intron 1, where there are numerous known splicing and regulatory elements [24]. Although the function of this SNP has not been characterised, our in silico analysis indicates higher *PD-1* expression in association with the *PDCD1*rs7421861 A allele and also supports aberrant *PD-1* splicing as a means by which the G allele protects against cancers. An alteration in the prevalence of *PD-1* transcripts, with the G allele suppressing PD-1 mediated immune exhaustion and protecting against the development of cancers—including viral hepatitis HCC—has been previously suggested [24,30,31,32,33]. The data from the Newcastle cohort were in keeping with this, with the wild type “A” allele being significantly more common in NAFLD patients with cancers. This was not so apparent in the smaller cohorts from Berne and Milan, with fewer non-cirrhotic cases and fewer women.

Possibly the most important take home message from this paper, is that while the genetic variations in the genes influencing the environment in which HCC develops may differ in populations, there are likely to be pathways that are shared. Although we did not confirm these specific *PDCD1* SNP associations independently in the smaller replication cohorts from Berne and Milan, our study was limited to two SNPs and does not rule out a key role for dysregulation of PD-1 in these or other populations, at different levels as yet undefined. While the NAFLD predisposing genes promoting progression of fatty liver disease have been demonstrated to play a key role in NAFLD-HCC development, mainly mediated through the induction of steatohepatitis and fibrogenesis, genes regulating immunoregulatory pathways may have a role at least equally important—more so in those cases without significant fibrosis. This study highlights a potential role for PD-1 in susceptibility to HCC in NAFLD which is worthy of further study—especially as PD-1 is a druggable target. Furthermore, for patients with NAFLD-HCC, histological subtyping may carry information that will be important to define as we enter an era of immuno-oncology therapies. Going forward, particularly as emerging data suggest that those with NAFLD-HCC may be less likely to respond to agents inhibiting PD-1/PD-L1 [61], understanding the immune environment is essential, as is identifying clinically useful tissues and blood-based biomarkers. Immunoregulatory SNPs have in the past directed liver disease-related therapies [62]. Those SNPs characterised here, or others, may yet have roles in stratifying surveillance or therapies for NAFLD-HCC. Their major contribution, however, may well be their highlighting key immunoregulatory mechanisms for further study.

## 5. Conclusions

We report that the genetic predisposition to HCC development in NAFLD may be influenced by immunoregulatory genes—most notably in this study, *PDCD1*—in addition to genes predisposing to increased fat accumulation and progression to advanced fibrosis and cirrhosis.

## Figures and Tables

**Figure 1 cancers-13-01412-f001:**
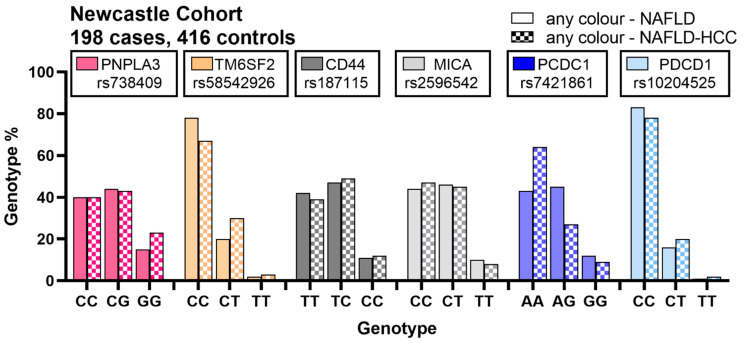
Allele frequencies of candidate SNPs in Newcastle cohorts.

**Figure 2 cancers-13-01412-f002:**
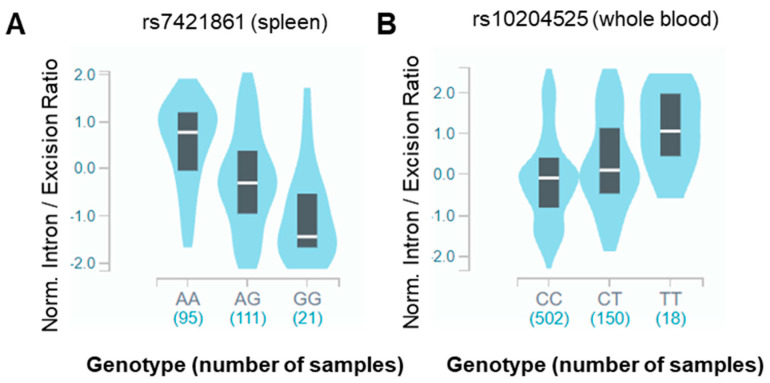
The figure shows sQTL violin plots for *PDCD1* rs7421861 on LINC01238 (intron location Chr2:241971462-241971680, hg38) in spleen (**A**) and *PDCD1* rs10204525 for LINC01237 (intron location Chr2: 241882635-241887744, hg38) in whole blood (**B**).

**Table 1 cancers-13-01412-t001:** Demographic characteristics of Newcastle NAFLD and NAFLD-HCC cohorts.

Phenotype	Group	NAFLD (*n* = 416)	NAFLD/HCC (*n* = 198)	*p*-Value ^1^
Age (Mean ± SD)		52.97 ± 0.58	72.21 ± 0.65	<0.0001
Gender	male (%) female (%)	232 (55.8) 184 (42.8)	157 (79.3) 41 (20.7)	<0.0001
BMI (Mean ± SD)		35.03 ± 0.28	31.96 ± 0.44	<0.0001
Diabetes	no (%) yes (%)	198 (48.1) 214 (51.9)	60 (30.3) 138 (69.7)	<0.0001
Cirrhosis	no (%) yes (%)	353 (84.9) 63 (15.1)	77 (38.9) 121 (61.1)	<0.0001

^1^*p*-values estimated by Mann–Whitney or Chi Square tests for continuous or categorical datasets respectively.

**Table 2 cancers-13-01412-t002:** Allelic analyses, including multivariate regression and conditioning on the fat-regulatory genes.

Gene		Identity	*p* Value	OR	Conditioned on *PNPLA3 + TM6SF2*	
					*p* Value	OR
*PNPLA3*	C > G	rs738409	0.01750 *	1.33 (1.05–1.68)	NA	NA
*TM6SF2*	C > T	rs58542926	0.00531 **	1.60 (1.15–2.22)	NA	NA
*MICA*	T > C	rs2596542	0.37940	0.89 (0.68–1.16)	0.3741	0.89 (0.68–1.16)
*CD44*	C > T	rs187115	0.42710	1.11 (0.86–1.44)	0.4155	1.11 (0.86–1.45)
*PDCD1*	A > G	rs7421861	0.00014 ***	0.59 (0.45–0.78)	0.000465 ***	0.61 (0.47–0.81)
*PDCD1*	C > T	rs10204525	0.16470	1.31 (0.90–1.91)	0.1838	1.29 (0.88–1.91)
Age, Sex, Cirrhosis, T2DM						
*PNPLA3*	C > G	rs738409	0.06432	1.49 (0.98–2.26)	NA	NA
*TM6SF2*	C > T	rs58542926	0.48770	0.82 (0.47–1.44)	NA	NA
*MICA*	T > C	rs2596542	0.84010	0.95 (0.60–1.52)	0.9079	0.97 (0.61–1.56)
*CD44*	C > T	rs187115	0.37700	1.23 (0.77–1.96)	0.3132	1.27 (0.80–2.04)
*PDCD1*	A > G	rs7421861	0.00152 *	0.49 (0.31–0.76)	0.001514 **	0.49 (0.31–0.76)
*PDCD1*	C > T	rs10204525	0.02212 *	2.11 (1.11–3.99)	0.007101 **	2.49 (1.28–4.86)

*p*-values * < 0.05; ** < 0.01; *** < 0.001.

**Table 3 cancers-13-01412-t003:** Genotype distribution between cirrhotic and non-cirrhotic (NC) NAFLD and NAFLD-HCC patients.

		NAFLD Control			NAFLD-HCC		
		Total (%)	NC (%)	Cirrhotic (%)	Total (%)	NC (%)	Cirrhotic (%)
*PNPLA3* rs738409	CC	170 (40.9)	156 (44.2)	14 (22.2)	67 (33.84)	35 (46.1)	32 (26.2)
	CG	184 (44.2)	150 (42.5)	34 (54.0)	85 (42.93)	29 (38.2)	56 (45.9)
	GG	62 (14.9)	47 (13.3)	15 (23.8)	46 (23.23)	12 (15.8)	34 (27.9)
*TM6SF2* rs58542926	CC	323 (77.6)	281 (79.6)	42 (66.7)	132 (66.7)	57 (74)	75 (62.0)
	CT	85 (20.4)	67 (19)	18 (28.6)	60 (30.3)	19 (24.7)	41 (33.9)
	TT	8 (2)	5 (1.4)	3 (4.8)	6 (3)	1 (1.3)	5 (4.1)
CD44 rs187115	TT	175 (42.1)	150 (42.5)	25 (39.7)	79 (39.9)	27 (35.1)	52 (43)
	CT	197 (47.3)	165 (46.7)	32 (50.8)	96 (48.5)	40 (51.9)	56 (46.3)
	CC	44 (10.6)	38 (10.8)	6 (9.5)	23 (11.6)	10 (13)	13 (10.7)
*MICA* rs2596542	CC	183 (44)	159 (45)	24 (38.1)	92 (46.9)	31 (41.3)	61 (50.4)
	CT	190 (45.7)	158 (44.8)	32 (50.8)	89 (45.4)	37 (49.3)	52 (43)
	TT	43 (10.3)	36 (10.2)	7 (11.1)	15 (7.7)	7 (9.3)	8 (6.6)
*PDCD1* rs7421861	AA	180 (43.3)	155 (43.9)	25 (39.7)	126 (63.6)	50 (64.9)	76 (62.8)
	AG	189 (45.4)	159 (45)	30 (47.6)	53 (26.8)	22 (28.6)	31 (25.6)
	GG	47 (11.3)	39 (11)	8 (12.7)	19 (9.6)	5 (6.5)	14 (11.6)
*PDCD1* rs10204525	CC	345 (82.9)	289 (81.9)	56 (88.9)	154 (78.6)	56 (73.7)	98 (81.7)
	CT	66 (15.9)	59 (16.7)	7 (11.1)	38 (19.4)	18 (23.7)	20 (16.7)
	TT	5 (1.2)	5 (1.4)	0 (0)	4 (2)	2 (2.6)	2 (1.7)

**Table 4 cancers-13-01412-t004:** Demographic characteristics of Berne and Milan NAFLD and NAFLD-HCC cohorts.

		Newcastle Cohort			Berne Cohort			Milan Cohort *	
Phenotype		NAFLD (*n* = 416)	NAFLD-HCC (*n* = 198)	NAFLD (*n* = 76)	NAFLD-HCC (*n* = 84)	*p* Value	NAFLD (*n* = 102)	NAFLD-HCC (*n* = 109)	*p* Value
Age (Mean ± SD)		53.0 ± 0.6	72.2 ± 0.7	53.7 ± 1.3	66.7 ± 0.9	<0.0001	63.7 ± 1.1	66.9 ± 0.8	0.074
Gender (%)	male female	232 (55.8) 184 (42.8)	157 (79.3) 41 (20.7)	46 (60.5) 30 (39.5)	80 (95.2) 4 (4.8)	<0.0001	47 (52.2) 43 (47.8)	88 (81.5) 20 (18.5)	<0.0001
BMI (Mean ± SD)		35.0 ± 0.3	32.0 ± 0.4	32.7 ± 0.6	29.6 ± 0.6	<0.0001	30.0 ± 0.5	29.4 ± 0.5	0.445
Diabetes (%)	no yes	198 (48.1) 214 (51.9)	60 (30.3) 138 (69.7)	35 (46.1) 41 (53.9)	47 (56.0) 37 (44.0)	0.211	34 (40.5) 50 (59.5)	45 (44.6) 56 (55.4)	0.577
Cirrhosis (%)	no yes	353 (84.9) 63 (15.1)	77 (38.9) 121 (61.1)	55 (72.4) 21 (27.6)	15 (17.9) 69 (82.1)	<0.0001	6 (7.0) 80 (93.0)	16 (15.5) 87 (84.5)	0.068

*p*-values estimated by Mann–Whitney or Chi Square tests for continuous or categorical data sets, respectively. * some categorical data unavailable.

**Table 5 cancers-13-01412-t005:** The demographic characteristics of the combined European cohort.

Combined Cohorts	Group	NAFLD (*n* = 594)	NAFLD-HCC (*n* = 391)	*p* Value
Age (Mean ± SD)		54.74 ± 0.50	69.52 ± 0.47	<0.0001
Gender	male (%) female (%)	325 (55.8) 257 (44.2)	325(83.1) 66 (16.9)	<0.0001
BMI (Mean ± SD)		33.99 ± 0.25	30.78 ± 0.30	<0.0001
Diabetes	no (%) yes (%)	267 (46.7) 305 (53.3)	152 (39.6) 232 (60.4)	<0.030
Cirrhosis	no (%) yes (%)	414 (71.6) 164 (28.4)	108 (28.0) 278 (72.0)	<0.0001

**Table 6 cancers-13-01412-t006:** META analyses in the combined European cohort.

Gene	rs Identity	*p* Value	OR	Condition on PNPLA3+TM6SF2	
PNPLA3	rs738409	0.043946	1.20 (1.00–1.43)	NA	NA
TM6SF2	rs58542926	0.018524	1.37 (1.05–1.77)	NA	NA
PDCD1	rs7421861	0.026279	0.79 (0.65–0.97)	0.044663	0.81
PDCD1	rs10204525	0.123195	1.30 (0.93–1.83)	0.137854	1.30
Age, Gender, Cirrhosis, T2DM					
PNPLA3	rs738409	0.231807	1.18 (0.90–1.55)	NA	NA
TM6SF2	rs58542926	0.655423	0.91 (0.61–1.36)	NA	NA
PDCD1	rs7421861	0.181639	0.82 (0.61–1.10)	0.172542	0.81 (0.61–1.10)
PDCD1	rs10204525	0.024180	1.90 (1.09–3.30)	0.009843	2.13 (1.20–3.80)

**Table 7 cancers-13-01412-t007:** The eQTL effect size and significance for PDCD1 rs7421861 and PD-1 expression in indicated cell types, with the variant significantly associated with suppressed PD-1 expression.

rs7421861		
*p*-Value (-log10)	Effect Size	Tissue/Cell Type
2.028825	−0.06357	Whole_Blood
2.737219	−0.10859	blood
1.413284	−0.21721	macrophage_naive
1.481194	−0.22847	monocyte_IAV
2.160692	−0.38254	monocyte_Pam3CSK4
2.072197	−0.39053	monocyte_LPS
2.878066	−0.45225	macrophage_Listeria
4.672717	−0.52111	monocyte_R848

**Table 8 cancers-13-01412-t008:** The eQTL effect size and significance for PDCD1 rs10204525 and PD-1 expression in indicated cell types, with the variant significantly associated with increased PD-1 expression.

rs10204525		
*p*-Value (-log10)	Effect Size	Tissue/Cell Type
8.853481	0.964331	monocyte_Pam3CSK4
8.408515	0.96187	monocyte_LPS
9.49804	0.944919	monocyte_naive
2.34563	0.916861	monocyte_CD16_naive
5.266115	0.738328	CD4_T-cell_anti-CD3-CD28
7.349027	0.735695	monocyte_R848
4.761555	0.591227	CD8_T-cell_anti-CD3-CD28
6.339877	0.581108	monocyte_IAV
1.772399	−0.08286	Whole_Blood
5.520671	−0.25549	blood

## Data Availability

EMBL-EBI. https://www.ebi.ac.uk/gwas/studies/GCST90011885 (accessed on 25 August 2020); https://www.ebi.ac.uk/gwas/studies/GCST010861 (accessed on 25 August 2020) for GWAS information.

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
