# Peer review of "A *PDCD1* Role in the Genetic Predisposition to NAFLD-HCC?"

_cancers, 2021, doi:10.3390/cancers13061412_

Round 1
Reviewer 1 Report
The presented paper aims to evaluate the possible role of PDCD1 in the genetic predisposition to NAFLD-HCC.
Authors reported in three different cohorts of NAFLD patients with or without HCC the contribution of genetic variation of PDCD1 gene. This result could be potentially very important and with different clinical effects for the clinical follow-up of these patients, who are also candidate to immune-oncological therapies.
Basically, the topic of this paper is of potential interest and clinically meaningful.
Major comments:
- The mayor concern is about the calculation of the sample size to evaluate the effect of the genetic variation in the risk of HCC development. At least, a priori statistical power for the analysis of the SNP in the cohorts should be performed.
- The proportion of patients with NAFLD and NAFLD-HCC is very different among the three cohorts. Could you explain these differences? The inclusion criteria for Bern and Milan were the same as for Newcastle?. Which was the time recruitment for patients from Bern and Milan, and how long was the duration of follow-up for HCC patients in Bern and Milan?
- The main topic of this paper is to evaluate the effect of the genetic variation in the development of HCC in NAFLD cirrhotic and cirrhotic patients. The first part of the results that analyze the modification of BMI in the different TNM stages of HCC and the effects of histological subtypes in the survival of patients with NAFLD-HCC are of interest, but it is another topic than the main focus of the study, and should be described separately
Minor comments:
- The presence of a single table describing the clinical characteristics of the three cohorts could be helpful to understand the differences and similarities among them.
- Patients with HCC attributed to NAFLD are: 199 in Newcastle Cohort, 109 in Milan Cohort, 84 in Bern Cohort: total number is 392 and not 391 as reported in the paper.
- Please check the English grammar on the line 64 to 66. Please check for minor spell errors (line 150: seprarately).
Author Response
Thankyou very much for reviewing our paper and providing very constructive feedback.
Major comments:
The mayor concern is about the calculation of the sample size to evaluate the effect of the genetic variation in the risk of HCC development. At least, a priori statistical power for the analysis of the SNP in the cohorts should be performed.
Thankyou very much for this. We have now performed a power calculations with our cohort sizes, to evaluate their power to evaluate the effect of genetic variation on HCC risk in NAFLD. This was using the ‘case-control for discrete traits’ option in GPC software (http://zzz.bwh.harvard.edu/gpc/). This is described in the methods (lines 164-174), with the details of it shown in Supplementary Table 2.
The proportion of patients with NAFLD and NAFLD-HCC is very different among the three cohorts. Could you explain these differences? The inclusion criteria for Bern and Milan were the same as for Newcastle?. Which was the time recruitment for patients from Bern and Milan, and how long was the duration of follow-up for HCC patients in Bern and Milan?
Thankyou. As you have surmised, the selection criteria were not the same. These may go some way to explaining the differences. Additional detail is now provided in the methods (128-134) and also discussed further in the results (385-399).
The main topic of this paper is to evaluate the effect of the genetic variation in the development of HCC in NAFLD cirrhotic and cirrhotic patients. The first part of the results that analyze the modification of BMI in the different TNM stages of HCC and the effects of histological subtypes in the survival of patients with NAFLD-HCC are of interest, but it is another topic than the main focus of the study, and should be described separately.
Thankyou. We have accepted this advice. We have removed the reference to survival associated with histological subtype in section 3.1.2 of the paper. We have removed Supplementary Figure 1. We have removed reference in the discussion to the importance histological subtype for guiding future management of patients with NAFLD, as this is another topic.
We have retained the detail on the clinical variables, highlighting the differences between the cirrhotic and non-cirrhotic cases. This is a paper which we hope will inform clinical research and these features are highly relevant to patient management and have been interpreted alongside the genotypes. This includes the analysis of histological subtypes. I hope this is acceptable.
Minor comments:
The presence of a single table describing the clinical characteristics of the three cohorts could be helpful to understand the differences and similarities among them.
Thankyou. We have revised Table 4, including each of the cohorts, to increase the ease of comparison.
Patients with HCC attributed to NAFLD are: 199 in Newcastle Cohort, 109 in Milan Cohort, 84 in Bern Cohort: total number is 392 and not 391 as reported in the paper.
Apologies for this and thankyou for pointing it out. In fact the error is the other way around – there were 198 in the Newcastle cohort, not 199, with the total remaining 391. This error has been corrected throughout the paper.
Please check the English grammar on the line 64 to 66: Thankyou – the sentence has been revised
Please check for minor spell errors (line 150: seprarately). Thankyou again – that too has been corrected
Reviewer 2 Report
Review
In this article, the authors identified novel SNPs associated with NAFLD-HCC in PDCD1 gene. Comparing 594 patients with NAFLD with 391 patients with NAFLD-HCC, SNPs of PNPLA rs738409, TM6Sf2 rs58542926, PDCD1 rs7421861 and PDCD1 rs10204525 were associated with NAFLD-HCC. Particularly, after adjusting for age, body mass index, diabetes and cirrhosis, only PDCD1 SNPs were significantly associated with the development of NAFLD-HCC. As PDCD1 encodes PD-1, variations of immunoregulatory gene may influence HCC development in patients with NAFLD.
Comments:
This study addresses very interesting points. However, the data shown in the paper is too preliminary to state that SNPs of rs7421861 and rs10204525 in PDCD1 were associated with HCC development in NAFLD because their significance was not validated in other two European cohorts (Berne and Milian). Sup Table 11 shows the proportion of AA, GA and GG in PDCD1 rs7421861, and that of CC, TC and TT in PDCD1 rs10204525 were not significantly different between NAFLD and NAFLD-HCC.
In addition, the functional significance of SNPs in PDCD1 is not clarified in the current study; just speculated that it might be associated with immunoregulation.
Author Response
Thankyou – the point is well made and we agree. Presently we have avoided any claim that the SNPs are causally associated with HCC development in NAFLD. Larger cohorts will be required, as will functional studies – which may ultimately conclude that the impact of PDCD1 SNPs is greater or lesser, or via another route in the majority of NAFLD-HCC. However we do feel it is important to shift the major focus of genetic studies from fat regulatory genes to immunoregulatory ones and that our data supports this.
We have altered the title, raising a role for PDCD1 as a question, rather than a statement of fact. I hope that this will be acceptable.
Reviewer 3 Report
None
Author Response
Thankyou very much
Round 2
Reviewer 2 Report
Comments:
I can understand that the main point the authors newly show is genetic variation in not only genes related to fat metabolism but immunoregulatory genes could be associated with NAFLD-HCC development. However, as I mentioned for the first version, it is too preliminary to confirm that SNPs in PDCD1 gene were associated with HCC development. The validation study in the independent large cohort is necessary to corroborate the authors’ claim. If it is difficult to design the new validation study, at least the functional change or the expression change caused by specific SNPs in PDCD1 gene should be clarified. The comparison analysis of the expression level of PDCD1 between the cases with or without the SNPs will be useful. RNA-seq data in public database is available for the examination.
Author Response
Thankyou very much for your suggestion. We have analysed publicly available data, exploring gene expression changes in association with the SNPs. We have included an additional results section (3.6, lines 421-500), with 2 additional Tables (7 and 8) in the paper, two further Tables in the supplementary information (13 and 14), and an additional figure (Figure 2).
We have referenced these data in our revised discussion (lines 557-558; 569-571; 577).
In addition, we have clarified in our final paragraph that we are not claiming that these SNPs in the PDCD1 gene are associated conclusively with NAFLD-HCC, given they have not been independently replicated, but highlighting the importance of studying PDCD1 and PD-1 in NAFLD-HCC pathogenesis.
Round 3
Reviewer 2 Report
The analysis of PD-1 expression in genetic variants using public datasets gives very useful information to this paper, although the significance of the SNPs was not validated in another cohort. I think the current version is acceptable for the publication.